# ASDSV: Multimodal Generation Made Efficient with Approximate Speculative Diffusion and Speculative Verification

**Kaijun Zhou, Xingyu Yan, Xingda Wei, Xijun Li, Jinyu Gu**[*]
School of Computer Science, Shanghai Jiao Tong University

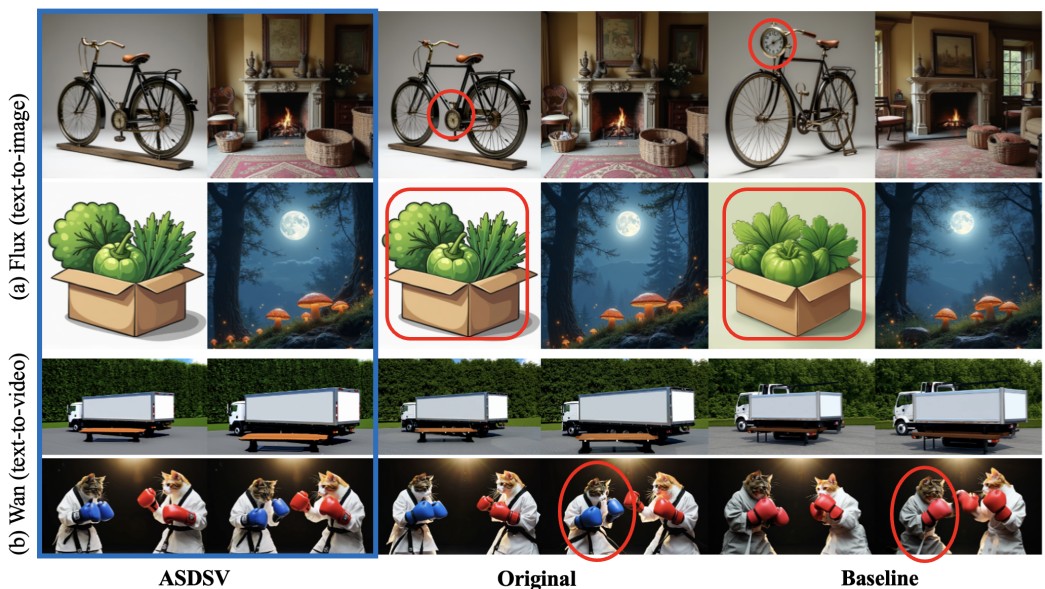

Figure 1: Visual comparisons of different methods on various generation tasks. From left to right: Approximate Speculative Diffusion with Speculative Verification (ASDSV), original model, Baseline (TeaCache). The red boxes highlight regions that differ significantly from the original image/video.

## Abstract

Diffusion in transformer is central to advances in high-quality multimodal generation but suffer from high inference latency due to their iterative nature. Inspired by speculative decoding's success in accelerating large language models, we propose *Approximate Speculative Diffusion with Speculative Verification (ASDSV)*, a novel method to enhance the efficiency of diffusion models. Adapting speculative execution to diffusion processes presents unique challenges.

First, the substantial computational cost of verifying numerous speculative steps for continuous, high-dimensional outputs makes traditional full verification prohibitively expensive. Second, determining the optimal number of speculative steps $K$ involves a trade-off between potential acceleration and verification success rates.

To address these, ASDSV introduces two key innovations: 1) A *speculative verification* technique, which leverages the observed temporal correlation between draft and target model outputs, efficiently validates $K$ speculative steps by only checking the alignment of the initial and final states, significantly reducing verification

---

[*]Corresponding author.

39th Conference on Neural Information Processing Systems (NeurIPS 2025).

overhead. 2) A *multi-stage speculative strategy* that adjusts $K$ according to the denoising phase—employing smaller $K$ during volatile early stages and larger $K$ during more stable later stages to optimize the balance between speed and quality. We apply ASDSV to state-of-the-art diffusion transformers, including Flux.1-dev for image generation and Wan2.1 for video generation. Extensive experiments demonstrate that ASDSV achieves up to $1.77\times$-$3.01\times$ speedup in model inference with a minimal 0.3%-0.4% drop in VBench score, showcasing its effectiveness in accelerating multimodal diffusion models without significant quality degradation. The code will be publicly available once the acceptance of the paper.

# 1 Introduction

Multimodal generation has achieved rapid progress in recent years, with diffusion models emerging as a dominant choice for producing high-quality, semantically coherent outputs across modalities such as images, videos, and text [7, 15]. As these models scale in capacity and complexity—exemplified by recent breakthroughs like Sora [3], Flux [18], and Wan2.1 [37]—their generative capabilities have reached unprecedented levels.

However, the inference latency of diffusion models remains a critical performance challenge due to the inherently iterative nature of the denoising process [1]. Each inference requires tens to hundreds of sequential denoising steps and each step is computationally expensive. This challenge is particularly pronounced in high-resolution image synthesis or long video generation, where the cumulative latency poses significant barriers to real-time applications and scalable deployment [41, 22].

To address this challenge, previous works have explored three primary directions: distillation, architectural modifications, and caching-based strategies. Distillation methods [29] compress pretrained models into faster variants but demand substantial computational resources for retraining and often degrade output quality. Architectural optimizations, such as attention compression and sparse attention mechanisms [43, 38], reduce per-step computation but also require retraining and are tightly coupled with model designs, limiting generalizability. Caching-based approaches [45, 42, 24] investigate the similarity between adjacent steps in the denoising process and propose skip-stepping strategies to reduce the number of denoising steps. Yet, this aggressive strategy may lead to a noticeable degradation in generation quality, as shown in Figure 1 (differences between Original and Baseline).

In language model inference, speculative decoding [19] has emerged as a powerful paradigm for latency reduction. It typically employs a lightweight draft model to predict $K$ tokens sequentially, which are then verified in parallel by the target model in a single inference batch, achieving speedups without compromising the output results. Inspired by its success, we explore adapting speculative decoding to diffusion models. A key observation motivates this direction: multimodal generation models often have corresponding draft models readily available, either as lightweight variants from the same family (e.g., Wan2.1-1.3B for Wan2.1-14B [37], Stable-Diffusion 3.5 medium and large variants [9] ) or through community-driven model compression efforts (e.g., Flux-lite for Flux.1-dev [18]).

However, adapting this strategy to diffusion models presents unique challenges. First, efficient verification is a significant hurdle. The continuous and high-dimensional nature of image and video data demands substantially more computational resources and memory for verification compared to discrete token sampling in language models. This can restrict batch sizes and has limited existing speculative approaches in diffusion [2] to low-resolution tasks such as CIFAR-10 [17]. To overcome this, we introduce a **approximate speculative verification** strategy. Instead of verifying all $K$ intermediate outputs from the draft model, our method only checks the initial and final speculative outputs. If both align with the target model's outputs within a defined tolerance, all intermediate steps are accepted. The insight for this efficient verification is that the outputs from draft and target models tend to exhibit similar variations across adjacent denoising steps (detailed in Sec3.1).

Second, determining the optimal number of speculative steps, $K$, involves a critical trade-off. A larger $K$ could theoretically yield greater acceleration, but it also increases the likelihood of the draft model's predictions diverging from the target model, leading to higher verification failure rates. To address this, we propose **multi-stage speculative strategy** that adopts multiple $K$ values based on the characteristic dynamics of diffusion models: early denoising steps exhibit rapid output changes, while later steps show gradual convergence [45, 10]. Consequently, our strategy employs a smaller

$K$ during the volatile early stages to minimize error accumulation and a larger $K$ in the more stable later stages to maximize performance.

In all, our proposed system, which we term *Approximate Speculative Diffusion with Speculative Verification (ASDSV)*, begins approximate speculative diffusion after letting the target model complete the first $N$ denoising steps. The initial denoising steps exhibit large variations, making them unsuitable for speculative execution. Then, ASDSV enters an interleaved execution pattern: the draft model executes $K$ steps, speculative verification is performed, and the target model executes a single step. This approach effectively reduces total computations while striving to preserve generation quality.

We apply ASDSV to a wide range of multimodal models, including two state-of-the-art diffusion transformers: Flux.1-dev [18] (image generation) and Wan2.1 [37] (video generation). Our extensive experiments demonstrate that ASDSV achieves up to $1.77\times$-$3.01\times$ speedup on model inference speed over the vanilla execution, while only incurring a minimal performance drop of 0.3%-0.4% in VBench score.

## 2 Related Work

### 2.1 Diffusion Model

Multimodal generation has achieved remarkable progress, with diffusion models continuously breaking state-of-the-art records [3, 12]. Diffusion models consist of an iterative process of denoising steps, where inference gradually removes noise from the initial data sample [15, 31]. By using attention mechanisms [34] and various optimization algorithm designs [25, 26, 23], diffusion in transformer (DiT) [30] models demonstrate excellent performance in text-to-image and text-to-video generation. State-of-the-art (SOTA) models have evolved from Pixart-Alpha (1.2B) [4] to Flux (12B) [18] in image generation; from Open-Sora (1.1B) [46] to Wan2.1 (14B) [37] and Step-Video (20B) [27] in video generation.

### 2.2 Diffusion Model Acceleration

**Distillation and Architectural Optimization**. Besides distillation methods [32, 6, 28] which requires substantial computational resources to re-train, many acceleration methods focus on optimizing the attention mechanism to reduce the computational cost. DistFastAtten [43] introduces spatial and temporal window attention with residual sharing, while Efficient-vDiT [8] and Sparse VideoGen [38] leverage the sparse properties of 3D full-attention in video generation to reduce the number of tokens involved in attention computation. Yet, such architectural optimizations either requires extra training or are tightly coupled with specific model designs.

**Cache**. Since the outputs of diffusion models after each denoising step exhibit strong similarity, cache-based methods have been proposed to reduce the number of denoising steps [42, 45, 24]. PAB [45] uses a step-skipping strategy (fixed-step) to skip some of the denoising steps. The blind step-skipping strategy may lead to performance degradation. TeaCache [24] designs a cache indicator to determine when to reuse the cached results (i.e., the output of the previous step) based on the similarity of the inputs of two adjacent steps, which avoids fixed-step skipping. However, directly reusing the cached outputs would still cause performance degradation, as shown in experiments.

**Parallelization**. Parallelization methods distributing computations across multiple GPUs, can also be effective for accelerating diffusion models (e.g., PipeFusion [11], DistriFusion [20], Classifier-Free Guidance Parallelism [10]) and orthogonal to our work.

### 2.3 Speculative Decoding

To reduce the latency of autoregressive language model inference, Speculative Decoding introduces a Draft-then-Verify decoding paradigm [39, 40, 19]. It needs to train a small model (draft model) for the large language model (target model). For decoding, it first efficiently generates multiple tokens by the draft model and then verifies all these tokens in parallel (in one batch) using the target model. Thereby, the target model inferences once but generates multiple tokens, which is faster than the original autoregressive decoding.

One prior work, Speculative Diffusion [2], make the first attempt to extend Speculative Decoding to diffusion models. However, it is only applied to low-resolution image-generation tasks (32×32) [17], as the exsiting verification process is too computationally expensive to be efficiently done for high-resolution tasks. Notably, the target model needs to generate all candidate steps in one batch for quickly verifying the draft steps, which is impractical for SOTA multimodal models [3, 37, 27].

# 3 Method

## 3.1 Approximate Speculative Diffusion with Speculative Verification

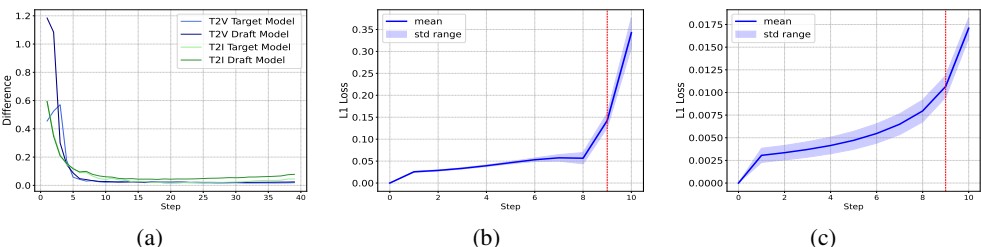

$$\text{(a)} \qquad\qquad \text{(b)} \qquad\qquad \text{(c)}$$

Figure 2: (a) shows strong temporal correlation between draft and target models in the diffusion process. (b) and (c) shows similar trends patterns of verification results: early steps show gradual followed by an abrupt divergence in the few last steps of the verification process.

Our design of ASDSV is motivated by two observations.

(1) The draft models for SOTA multimodal generation models are off-the-shelf, facillitating speculative diffusion. First, as multimodal generative models typically exhibit large parameter sizes and high computational demands, they are usually released with variants of varying sizes to accommodate diverse hardware capabilities. For examples, Wan2.1-1.3B for Wan2.1-14B and Stable-Diffusion 3.5's medium and large variants. Second, the community-developers also provide a rich pool of draft models through model compression techniques for resource-constrained environments.

(2) The output trends of draft and target models show temporal correlation. The *output trend* depicts that the differences between adjacent steps in the denoising process of a diffusion model, while difference between two steps is defined as L1 loss between the outputs of them.

$$\text{L1\_loss}(D_i, T_i) = \text{mean}(\|D_i - T_i\|)$$

$D_i$ and $T_i$ denote the predicted outputs from the draft and target models at step i, respectively. As shown in Figure 2(a), the output trends of draft and target models are close to each other. This refers to our key empirical observation: the strong temporal correlation between draft and target models in the diffusion process. Specifically, the magnitude of change of the draft model's output and the target model's output between consecutive steps is highly similar (i.e., $\|D_i - D_{i-1}\|_1 \approx \|T_i - T_{i-1}\|_1$) after the first $N$ initial steps. This observed characteristic holds for both text-to-image (T2I) and text-to-video (T2V) generation models.

For any adjacent $K$ denoising steps for the draft and target models, we mark their outputs as $D_N$, $D_{N+1}, ..., D_{N+K}$, and $T_N, T_{N+1}, ..., T_{N+K}$, respectively. Since the output trends of draft and target models are close to each other, if the differences (L1_loss) between $D_{N+1}$ and $T_{N+1}$ and between $D_{N+K}$ and $T_{N+K}$ are both smaller than a threshold, we can speculatively conclude that all the differences between intermediate $D_{N+i}$ and $T_{N+i}$ are also within the threshold. Thereby, unlike traditional speculative decoding that verifies every output ($T_{N+1}, ..., T_{N+K}$) of the draft model, we propose a *speculative verification* strategy which only verifies the output of the draft model at the beginning and end of the sequence ($T_{N+1}$ and $T_{N+K}$), which significantly reduces the verification cost for diffusion models. Here's a formalized expression for speculative verification:

$$D_{N+1} \leftarrow M_q(T_N), \quad D_{N+i} \leftarrow M_q(D_{N+i-1}), i = 2, \ldots, K$$

$$T_{N+1}, T_{N+K} \leftarrow M_p(D_{N+1}), M_p(D_{N+K}), \quad M_p \text{ and } M_q \text{ has strong temporal correlation}$$

$$\text{If L1\_loss}(D_{N+1}, T_{N+1}) \leq \delta \text{ and L1\_loss}(D_{N+K}, T_{N+K}) \leq \delta,$$

$$\text{then accept } D_{N+1}, \ldots, D_{N+K}, \text{ else reject and generate from } T_{N+2}$$

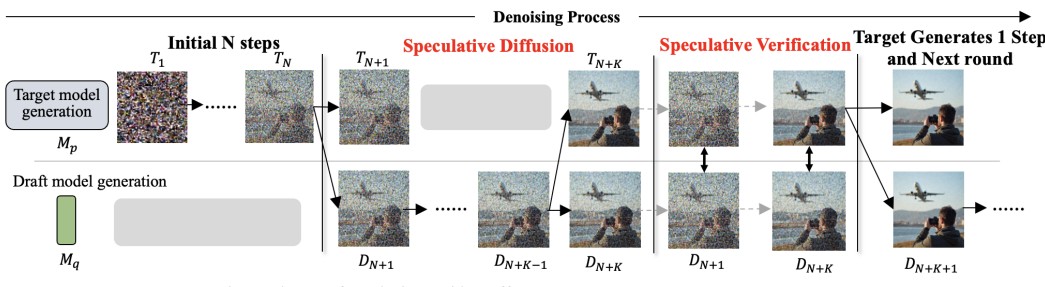

Figure 3: Overview of ASDSV compared with vanilla speculative diffusion. After $N$ target model steps, ASDSV sequentially samples $K$ steps within draft model, and the target model verifies the noises by comparing the first and last step's noises. In contrast, vanilla speculative diffusion verifies the noise of each step.

**Workflow.** Figure 3 illustrates the workflow of a denoising process with ASDSV. The initial $N$ steps are generated by the target model without any speculative steps, where the input for each step consists of the timestep embedding at the step and the output image of the previous step. After that, the repeated Speculative Diffusion and Speculative Verification process starts.

For each round of Speculative Diffusion, the draft model sequentially samples $K$ steps, where the initial step's input image comes from the target model's output of the previous round; the target model only executes for the first step and the last step (used for verification), where the last step's input image comes from the draft model's output ($D_{N+k}$). The following Speculative Verification will calculate two differences between $D_{N+1}$ and $T_{N+1}$, and between $D_{N+k}$ and $T_{N+k}$. The verification succeeds if both differences are smaller than a threshold (a hyper-parameter, $\delta$). If so, next round of Speculative Diffusion and Speculative Verification starts. Otherwise, the target model generates $T_{N+2}$ and then a new round starts.

**Batched Pipeline Optimization.** As shown in Figure 3, the target model needs to generate the $K$ step as the last step of the current round's verification process, and the $K + 1$ step as the first step of subsequent round. If the verification succeeds, $T_{N+K}$ is similar with $D_{N+K}$, so the target model can generate $T_{N+K+1}$ based on $D_{N+K}$. This allows for pipelining the current round's verification process and the next round's generation process in parallel. Thereby, ASDSV batches the draft sampling results $D_{N+K-1}$ and $D_{N+K}$ as the input of the target model, which then generates $T_{N+K}$ and $T_{N+K+1}$ in parallel.

## 3.2 Multi-Stage Speculative Strategy

According to Figure 2(a), we can also observe that during the denoising process, the differences between adjacent steps are big in the early stages and small in the later stages, which is an intrinsic property of diffusion models [45, 24]. If we choose a large $K$ value (speculative steps), the verification failure probability will be high in the early stages, leading to more wasted computation. In contrast, if we choose a small $K$ value, the verification failure probability will remain low in different stages, but the acceleration ratio will be low. Therefore, unlike traditional speculative decoding approaches that employ a fixed number of speculative steps $K$ throughout the entire decoding process, we propose a multi-stage speculative strategy which chooses different $K$ values for different stages during the denoising process.

Specifically, ASDSV uses a three-stage strategy. Stage-0: The initial N steps is the warmup stage which is used to establish the foundation for high-quality generation. Stage-1: The subsequent warmup stage employs a smaller $\gamma_1$ as $K$ value, to preserve a high success rate of verification during the early stage. Stage-2: The main generation stage employs a larger $\gamma_2$ as $K$ value, to achieve a higher acceleration ratio. Such a strategy can achieve an optimal balance between generation speed and generation quality.

We choose the hyperparameters based on the step-metric results in both image and video scenarios. We illustrate with a red vertical line in Figure 2 (b) and (c), the verification error curve exhibits a sharp

increase at the 9-th step. This inflection point shows that the draft model's outputs are accumulating errors and starting to diverge significantly from the target model's. Therefore, it represents the appropriate moment to halt the drafting phase. So we set $\gamma_2$ as 9 for Stage-2 to achieve both high success rate of verification and high acceleration ratio. More details about other stages are provided in the Appendix A.3.

### 3.3 Walltime Improvement

We denote the draft model as $M_q$ and the target model as $M_p$. Let $\gamma_1$ and $\gamma_2$ be the number of speculative steps in the Stage-1 and Stage-2, respectively. The running time of one step of $M_q$ is $T_{M_q}$ and that of $M_p$ is $T_{M_p}$. Let $c$, the *cost coefficient*, be the ratio of $T_{M_q}$ to $T_{M_p}$. The number of steps in Stage-0 is denoted as $N$. Assuming Stage-1 and Stage-2 respectively complete $\alpha_1$ and $\alpha_2$ speculative rounds and all verifications are successful, ASDSV can achieve a maximum speedup as follows.

$$N + \alpha_1\gamma_1 + \alpha_2\gamma_2 = \text{Total steps} \tag{1}$$

$$\text{Vanilla Diffusion: Total steps} \times T_{M_p} \tag{2}$$

$$\text{ASDSV: } NT_{M_p} + T_{M_q}(\text{Total steps} - N) + T_{M_p}(1 + \alpha_1 + \alpha_2) \tag{3}$$

Note that $M_p$ generates the last step of the current round and the first step of the next round in parallel for each round, except for the first step of the first round. Therefore, in ASDSV, the cost for $M_p$ is $NT_{M_p} + T_{M_p}(1 + \alpha_1 + \alpha_2)$. The theoretical upper bound of speedup is:

$$\frac{N + \alpha_1\gamma_1 + \alpha_2\gamma_2}{N + 1 + \alpha_1(c\gamma_1 + 1) + \alpha_2(c\gamma_2 + 1)} \tag{4}$$

We also provide the expected speedup of ASDSV and its proof in Appendix A.5.

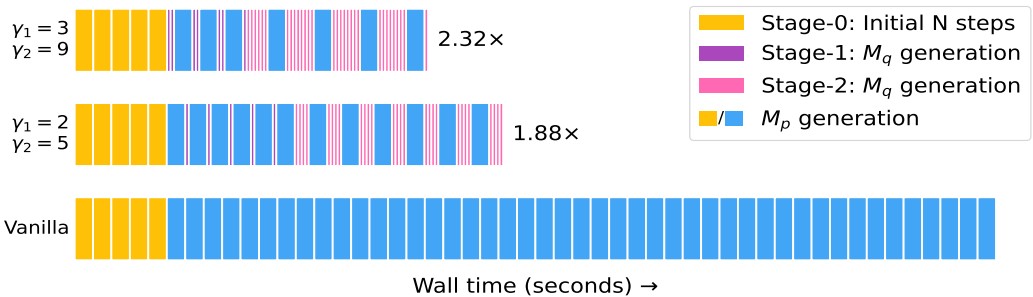

Figure 4: A simplified trace diagram to show ASDSV under different settings on Wan2.1 using 50 steps. The purple blocks and pink blocks represent steps where results are generated by the draft model in Stage-1 and Stage-2, respectively, $N$ is set to 10% of total steps. The first row shows ASDSV with $\gamma_1 = 3$ and $\gamma_2 = 9$, the second row shows ASDSV with $\gamma_1 = 2$ and $\gamma_2 = 5$, and the third row shows vanilla diffusion.

Figure 4 presents the concrete theoretical upper bound of speedup of ASDSV under different settings. However, since there can be verification failures in the speculative steps, the theoretical upper bound is hard to achieve. Furthermore, in our experiments, $c$ was 0.60 for text-to-image models and 0.19 for text-to-video models. In contrast, the cost coefficient $c$ is much smaller in large language models (often less than 0.05 [19]). As this work focuses on making speculative diffusion effective in SOTA multimodal generation models, the exploration of optimal draft model is orthogonal.

## 4 Experiments

### 4.1 Settings

**Models and Baselines.** We evaluate ASDSV on two representative diffusion models: Flux.1-dev [18] for image generation tasks and Wan2.1 [37] for video generation tasks. We compare ASDSV with two state-of-the-art studies on accelerating diffusion inference: Teacache [24] (cache-based method)

for both image and video generation, and Sparse VideoGen [38] (architectural optimization) for video generation acceleration.

**Metrics and Datasets.** We evaluate ASDSV and other baselines on two aspects: generation quality and generation efficiency. For image generation with Flux.1-dev, we evaluate the generation quality using Frechet Inception Distance (FID) [14], Inception Score (IS) [33], Learned Perceptual Image Patch Similarity (LPIPS) [44], Peak Signal-to-Noise Ratio (PSNR) [35], and CLIP score [13]. Each comparison method generates 10k images using the COCO Captions 2014 dataset [5]. FID, LPIPS and PSNR are metrics for similarity assessment, i.e., how different the generated images are from the ground truth (the output of the original model). CLIP score measures the semantic alignment between the input prompt and the generated images and IS measures the diversity of the generated images.

For video generation with Wan2.1, like prior studies [24], we evaluate the generation quality using VBench Score [16] to evaluate the visual quality of the generated videos, LPIPS, PSNR and Structural Similarity Index Measure (SSIM) [36] to evaluate the similarity between the generated videos and the ground truth. Each comparison method generates 2k videos using the VBench dataset [16].

For generation efficiency evaluation across both models, we use Floating Point Operations (FLOPs) and inference latency as metrics.

**Implementation Details.** We implement two variants of our method: ASDSV-fast and ASDSV-slow. The ASDSV-slow variant performs the speculative verification and samller $K$ for the Stage-1 speculative step to achieve better visual quality. The ASDSV-fast variant skips the verification entirely, assuming the draft samples pass the verification, and uses the same $K$ for the Stage-1 and Stage-2 speculative steps to achieve higher efficiency, which is a more aggressive speculative diffusion strategy.

For both image and video generation, we set the total number of denoising steps to 50. For text-to-image generation, we use Flux-SVD[21] as the draft model. For ASDSV-slow, we set $\gamma_1 = 3$, $\gamma_2 = 9$, and warmup ratio to 15%. For ASDSV-fast, we use $\gamma_1 = \gamma_2 = 9$. We set initial steps $N$ to 8% of total steps for both variants and a verification threshold ($\delta$) of 0.02. For text-to-video generation, we use Wan2.1-1.3B as the draft model. For ASDSV-slow, we set $\gamma_1 = 2$, $\gamma_2 = 9$, and warmup ratio to 25%. For ASDSV-fast, we use $\gamma_1 = \gamma_2 = 9$. We set initial steps $N$ to 10% of total steps for the fast variant and 15% for the slow variant with ($\delta$) 0.2.

As for the baselines, we use the same settings as in the original papers: Teacache [24] threshold is 0.25 for slow and 0.6 for fast in text-to-image generation, and 0.14 for slow and 0.2 for fast in text-to-video generation. Sparse VideoGen [38] is used with sparsity 0.10 for fast and 0.20 for slow, without enabling custom kernel and FP8 quantization. We measure the latency per sample on a single NVIDIA A800 GPU using Pytorch 2.6.0 and CUDA 12.4.

## 4.2 Main Results

**Comparison with Draft Models.**  Besides the above-mentioned baselines, we also compare ASDSV with using the different draft models for multimodal generation, which can be regarded as distillation-based methods. As provided in the Appendix A.1, the result is that ASDSV consistently outperforms all draft models, which also shows the effectiveness and broad applicability of ASDSV across diverse compression techniques.

**Results of Evaluation Metrics.**  Table 1 and 2 present the quantitative evaluation of efficiency and visual quality using the evaluation metrics described in Section 4.1. Compared to both cache-based and architectural optimization methods, ASDSV delivers better visual quality and enhanced generation efficiency across different modalities, base models, resolutions and video length.

In Table 1 (text-to-image), ASDSV-slow achieves the best visual similarity metrics (FID, LPIPS, PSNR) and attaining nearly $1.3\times$ speedup. Although ASDSV-fast achieves the lowest FLOPs count, it doesn't attain the minimum latency. This discrepancy arises because the quantized draft model doesn't reach optimal acceleration on A800 GPUs. On hardware supporting efficient low-precision computation (e.g., 50-series GPUs), our approach would likely achieve ideal acceleration, fully realizing its theoretical efficiency advantages.

Table 1: Efficiency and visual quality comparison of ASDSV and other baselines on image generation. Flux-SVD is the draft model in ASDSV.

| Method | Efficiency | | | Visual Quality | | | | |
|---|---|---|---|---|---|---|---|---|
| | FLOPs (T) ↓ | Latency (s) ↓ | Speedup ↑ | FID ↓ | CLIP ↑ | LPIPS ↓ | IS ↑ | PSNR ↑ |
| **FLUX.1-dev (512×512)** | | | | | | | | |
| FLUX.1-dev ($T = 50$) | 1082.5 | 9.18 | 1× | - | 31.14 | - | 26.32 | - |
| TeaCache-fast | 330.4 | **4.25** | 2.16× | 8.16 | **31.34** | 0.30 | 25.66 | 28.89 |
| TeaCache-slow | 566.8 | 5.73 | 1.6× | 4.33 | 31.20 | 0.14 | 26.7 | 31.67 |
| ASDSV-fast | **206.3** | 6.25 | 1.47× | 4.11 | 31.15 | 0.11 | 26.66 | 32.95 |
| ASDSV-slow | 292.3 | 7.3 | 1.26× | **3.90** | 31.15 | **0.10** | 26.52 | **33.14** |

Despite not achieving the highest IS and CLIP scores, ASDSV maintains scores closest to those of the original model, which aligns with our primary goal of achieving higher fidelity to the original model's output as the ground truth.

Table 2: Efficiency and visual quality comparison of ASDSV and other baselines on video generation.

| Method | Efficiency | | | Visual Quality | | | |
|---|---|---|---|---|---|---|---|
| | FLOPs (P) ↓ | Latency (s) ↓ | Speedup ↑ | VBench ↑ | LPIPS ↓ | PNSR ↑ | SSIM ↑ |
| **Wan2.1 (81 frames, 832×480)** | | | | | | | |
| Wan2.1 ($T = 50$) | 168.1 | 924 | 1× | 82.69 | - | - | - |
| Sparse-fast | 138.4 | 761 | 1.21× | 78.97 | 0.75 | 8.6 | 0.27 |
| Sparse-slow | 153.1 | 789 | 1.17× | 81.91 | 0.70 | 9.2 | 0.30 |
| TeaCache-fast | 84.2 | 462 | 2× | 81.83 | 0.45 | 14.2 | 0.49 |
| TeaCache-slow | 114.4 | 628 | 1.47× | 82.17 | 0.37 | 15.75 | 0.55 |
| ASDSV-fast | **52.6** | **307** | **3.01×** | **82.36** | 0.41 | 14.89 | 0.51 |
| ASDSV-slow | 74.4 | 521 | 1.77× | 82.29 | **0.33** | **16.58** | **0.58** |

In Table 2 (text-to-video), ASDSV-fast achieves the highest 3.01× speedup with better VBench score compared to TeaCache-slow. Additionally, ASDSV-slow achieves the most faithful reproduction (0.3%-0.4% VBench score degradation), along with higher acceleration 1.77× than TeaCache-slow. These results demonstrate that ASDSV not only preserves the better enerative performance of the models but also achieves greater acceleration than existing methods especially in text-to-video generation.

**Visualization.** Figure 1 presents the images and the videos (the first frame and last frame of the video) generated by ASDSV compared with those by Teacache and the vanilla. Experimental results show that ASDSV not only produces higher visual quality, but also achieves better alignment with the ground truth (original images and videos). Specifically, our method better preserves the color fidelity, positioning and details of characters compared with Teacache. Additional visual comparisons can be found in the Appendix D.

## 4.3   Ablation Study

**Verification Strategy Comparison.** We configure ASDSV-medium with $\gamma_1 = 3$ and $\gamma_2 = 9$, using a warmup ratio of 20%. As shown in Figure 6, we design three verification schemes to compare the trade-off between efficiency and quality. Our method offers multiple configurable variants, allowing users to select the optimal configuration based on their specific requirements.

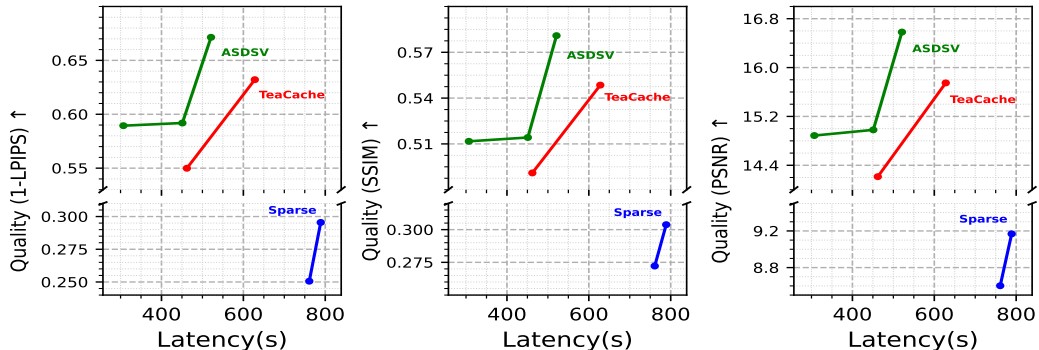

Figure 5: Quality-latency comparison of text-to-video diffusion models. The proposed ASDSV method demonstrates superior visual fidelity while maintaining lower inference latency compared to TeaCache[24] and Sparse VideoGen[38], evaluated under the Wan2.1 model.

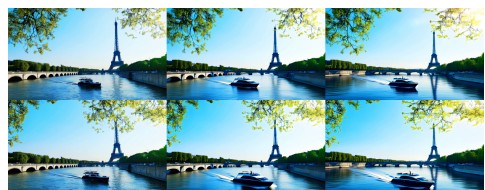

(a) prompt: A boat sailing leisurely along the Seine River with the Eiffel Tower in background, surrealism style.

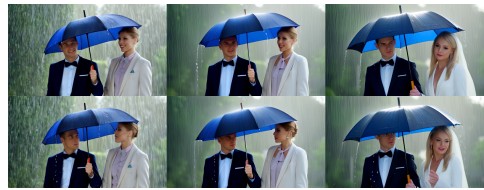

(b) prompt: A couple in formal evening wear going home get caught in a heavy downpour with umbrellas, zoom in.

Figure 6: Visual comparisons of different verification strategies on text-to-video generation. From left to right: ASDSV-fast, ASDSV-medium, and ASDSV-slow. Each column presents the first and last frames of the generated video.

**Performance at different Resolution and Length.** Figure 7 illustrates that ASDSV maintains robust acceleration performance across varying video lengths and resolutions. This sustained efficiency highlights ASDSV's capacity to expedite sampling for extended and high-fidelity video content, thereby addressing practical and effective requirements in more complex applications.

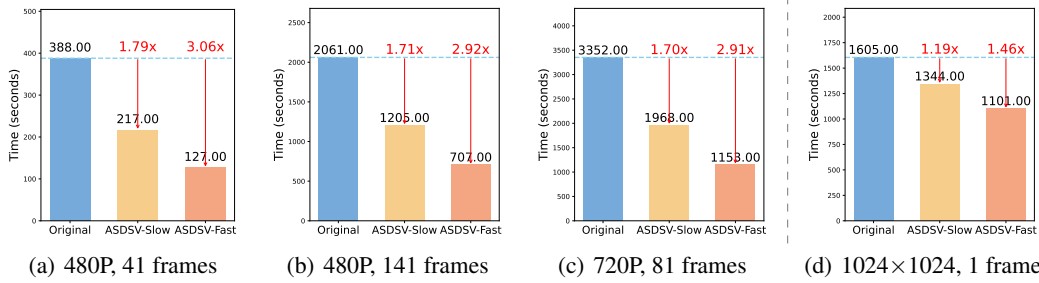

(a) 480P, 41 frames    (b) 480P, 141 frames    (c) 720P, 81 frames    (d) 1024×1024, 1 frame

Figure 7: Inference efficiency at different video lengths and resolutions. (a), (b), and (c) show different resolutions and video lengths using Wan2.1, while (d) corresponds to different resolutions using Flux.1-dev.

**Sensitivity Analysis and Tuning Process for $\gamma$.** Both $\gamma_1$, $\gamma_2$ depend on the temporal correlation between draft and target model on different stages, so ASDSV is sensitive to these parameters. Nevertheless, since the temporal correlation can be profiled offline, we can systematically search for the best candidates for these hyperparameters for each model offline. We generated 200 videos per setting for this specific study on Wan2.1 model, following are the detailed analysis and tuning methods of hyperparameters $\gamma$.

**Analysis of $\gamma_1$.** First, we use brute force to search $\gamma_1$ because the space is small: it can only be an integer in (1,5). Below is the search space of Wan2.1. We only present one model result for simplicity, other models are the same.

Table 3: Ablation study on the choice of speculative step $\gamma_1$ in Stage-1.

| $\gamma_1$ | LPIPS↓ | PSNR↑ | SSIM↑ | Speedup |
|---|---|---|---|---|
| 2 | 0.23 | 18.57 | 0.61 | 1.70x |
| 3 | 0.30 | 16.80 | 0.59 | 1.71x |
| 4 | 0.30 | 16.76 | 0.59 | 1.72x |

**Analysis of $\gamma_2$.** Because the search space of $\gamma_2$ is larger than $\gamma_1$ (an integer selected from range (1,15)), we developed a two-step iterative-guided search process for $\gamma_2$ to balance speed-fidelity trade-off: (1) Profile Model Dynamics: We conduct a small-scale sampling run to profile the model's behavior, generating plots similar to Figure 2. This allows us to observe the output change dynamics and step-wise verification loss, which are crucial for setting the $\gamma_2$ and decide the verfication threshold. (2) Iteratively Tune $\gamma_2$ and $\delta$: We start with a candidate $\gamma_2$ based on the theoretical speedup ceiling for our target. Next, using the verification loss curve (like Figure 2(b)(c)), we set a corresponding threshold $\delta$ above the observed loss at the $\gamma_2$-th step. We then incrementally increase $\gamma_2$ and adjust $\delta$, evaluating the trade-off on a validation set until the desired speedup is met with minimal quality impact. The following results show typical search results on Wan2.1 model, others are the same:

Table 4: Ablation study on the choice of speculative step $\gamma_2$ and verification threshold $\delta$ in Stage-2.

| $\gamma_2$ | $\delta$ | LPIPS↓ | PSNR↑ | SSIM↑ | Speedup |
|---|---|---|---|---|---|
| 3 | 0.1 | 0.24 | 18.89 | 0.60 | 1.2x |
| 9 | 0.2 | 0.23 | 18.57 | 0.61 | 1.7x |
| 12 | $\infty$ | 0.25 | 18.47 | 0.60 | 3.1x |

The baseline for these comparisons is the original model, corresponding to the setting $\gamma_1 = 0, \gamma_2 = 0$. The setting of $\delta = \inf$ represents an "always-accept" strategy that effectively bypasses the verification stage to achieve maximum acceleration, corresponding to our ASDSV-fast variant in Figure 5. We have summarized the tuning process described above into an automated method, which is detailed in Appendix B.

**Quality-Efficiency Tradeoff.** Figure 5 shows the trade-off between the quality and the efficiency of our method compared with other baselines. Notably, our method achieves the best scores not only on the reference-free VBench, but also on similarity metrics. This performance stems from the fact that ASDSV most closely aligns with the outputs of the original model, resulting in quality scores that are nearly indistinguishable from those of SOTA diffusion models.

## 5 Conclusion and Future Work

This paper introduces Approximate Speculative Diffusion with Speculative Verification (ASDSV), a novel speculative decoding method for multimodal generation. Different from traditional speculative decoding, ASDSV employs a speculative verification strategy specifically for diffusion models to minimize verification cost, by leveraging the temporal correlation between draft and target models. Meanwhile, we propose a multi-stage speculative strategy based on the dynamics of diffusion process to to balance the trade-off between generation speedup and performance. Experiments demonstrate that ASDSV achieves significant inference acceleration while maintaining high similarity and robust performance in both video and image generation models.

**Limitations and Future Work.** First, ASDSV performance depends on the cost coefficient between models, with limited gains when target models are already optimized or draft models lack efficiency. Second, our current verification failure handling discards all subsequent samples. Advanced recovery strategies could potentially enhance performance, such as employing binary search techniques to efficiently identify the last acceptable draft sample. Third, integration with parallelism strategies requires further research to maximize performance in distributed environments.

## Acknowledgments

We sincerely thank the anonymous reviewers, whose reviews, feedback, and suggestions have significantly strengthened our work. This research was supported in part by New Generation Information Technology Program from Shanghai Committee of Science and Technology (NO.25511104100), National Natural Science Foundation of China (No. 62432010), and the Fundamental Research Funds for the Central Universities.

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

# A Additional Experiments

## A.1 Results of Draft Model Comparison

In autoregressive language models, draft models ($M_q$) are typically selected from existing off-the-shelf smaller transformers with the same architecture as the target model ($M_p$). For example, T5-XXL (11B) as $M_p$ and T5-large (800M) as $M_q$. Following this practice, we evaluate the performance of different draft models on text-to-image generation. Flux.1-dev is the state-of-the-art text-to-image model, and Flux-lite (distilled model) and Flux-SVD (quantized model) are two efficient draft variants of Flux.1-dev.

Table 5 illustrates that ASDSV consistently outperforms both draft models, which demonstrates the effectiveness and broad applicability of ASDSV across diverse compression techniques. The quantized model Flux-SVD shows the best overall performance and is chosen as the draft model for Flux.1-dev in our experiments. FID, LPIPS and PSNR are reference-based metrics for measuring the similarity between generated results and Flux.1-dev original outputs. CLIP score measures the semantic alignment between the input prompt and the generated images and IS measures the diversity of the generated images.

Table 5: Performance of FLUX.1-dev compared with different draft models. *ASDSV w/ Flux-lite* and *ASDSV w/ Flux-SVD* are ASDSV that use Flux.1-dev as the target model and Flux-lite and Flux-SVD as draft model, respectively.

| Model | FLUX.1-dev | | | | |
|---|---|---|---|---|---|
| Score | FID ↓ | CLIP ↑ | LPIPS ↓ | IS ↑ | PSNR ↑ |
| Flux.1-dev | — | 31.14 | — | 26.32 | — |
| Flux-lite | 9.51 | 31.07 | 0.40 | 25.59 | 29.73 |
| Flux-SVD | 5.52 | **31.18** | 0.19 | 26.26 | 30.53 |
| ASDSV w/ Flux-lite | 6.62 | 31.20 | 0.22 | 25.68 | 30.49 |
| ASDSV w/ Flux-SVD | **4.97** | **31.24** | **0.11** | **26.36** | **32.69** |

Additionally, we improved the draft model baseline by combined the Stage-0 into the draft model diffusion process. ASDSV-slow and ASDSV-fast are the two variants of ASDSV with different settings, detailed in Experiments Settings of the main paper. Specifically, we use the target model to generate the first $N = 5$ steps as the draft model's initial steps (the same as the ASDSV-fast), and then use the draft model to generate the remaining steps. The evaluation metric results are shown in Table 6. The visual results are shown in Figure 8. ASDSV outperforms the Improved-Draft baseline in both visual quality metric (VBench score) and similarity metrics (LPIPS, PSNR, and SSIM).

Table 6: Efficiency and visual quality comparison of ASDSV and improved draft model baseline on text-to-video generation.

| Method | Efficiency | | | Visual Quality | | | |
|---|---|---|---|---|---|---|---|
| | FLOPs (P) ↓ | Latency (s) ↓ | Speedup ↑ | VBench ↑ | LPIPS ↓ | PNSR ↑ | SSIM ↑ |
| Improved-Draft | 28.0 | 256 | 3.67× | 82.23 | 0.42 | 14.70 | 0.50 |
| ASDSV-fast | 52.6 | 307 | 3.01× | **82.36** | 0.41 | 14.89 | 0.51 |
| ASDSV-slow | 74.4 | 521 | 1.77× | 82.29 | **0.33** | **16.58** | **0.58** |

## A.2 Multi-Seed Stability Analysis

Our method is insensitive to the random seed. This is because the seed in the diffusion process only affects the initial noise, whereas the efficacy of ASDSV depends on the temporal correlation between the draft and target models (as shown in Figure 2).

We conducted a new set of focused experiments on three additional seeds (3461, 772190542, 118270042274490399) for our main comparisons (Original vs. ASDSV). For this analysis, we

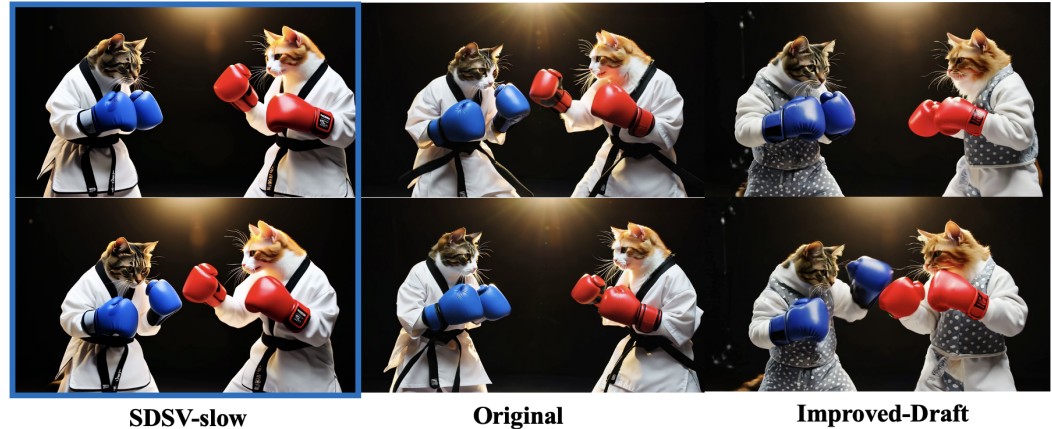

| SDSV-slow | Original | Improved-Draft |

Figure 8: Visual comparisons of improved draft model baseline on text-to-video generation.

generated 2000 images per seed for the Text-to-Image task and 200 videos per seed for the Text-to-Video task. Performance was consistent across all seeds (ASDSV achieves better performance than other acceleration methods). The standard deviations over multiple seeds results on Text-to-Image(T2I) over are shown in Table 7.

Table 7: Standard deviations over multiple seeds for T2I generation.

| Images Metric | Ratio (Original / ASDSV) |
|---------------|--------------------------|
| CLIP | $100.04\% \pm 0.10\%$ |
| IS | $103.84\% \pm 3.7\%$ |

Since most of the Text-to-Video(T2V) metrics are reference-based, we cannot calculate a relative ratio (original/accelerated method). Therefore, we used a comparison between the accelerated methods (ASDSV vs. Teacache) as shown in Table 8, and ASDSV is about 1.5x faster than Teacache.

Table 8: Standard deviations over multiple seeds for T2V generation.

| Videos Metric | Ratio (ASDSV / Teacache) |
|---------------|--------------------------|
| LPIPS$\downarrow$ | $88.02\% \pm 2.85\%$ |
| PSNR$\uparrow$ | $106.10\% \pm 1.25\%$ |
| SSIM$\uparrow$ | $104.95\% \pm 1.84\%$ |

### A.3 Analysis of Multi-Stage Speculative Strategy

ASDSV employs a multi-stage speculative strategy to balance the trade-off between generation speedup and performance: first N initial steps (Stage-0) combined with the subsequent Stage-1 serve as the warmup period for the diffusion process, establishing the foundation for high-quality generation, followed by the main generation stage as Stage-2. A key characteristic of the warmup period is the relatively large variation between outputs of adjacent steps, unlike the highly similar outputs observed between consecutive steps in Stage-2. To ensure higher quality model outputs, modifications to the architecture are typically avoided during the warmup period.

We evaluated two warmup approaches: the 2 steps method and a more conservative fixed ratio of 20% of total diffusion steps. After completing their respective warmup phases, they employ the same speculative strategies as ASDSV in Stage-2. Our experimental results in Figure 9 reveal a clear visual difference. Configuration (a) with minimal warmup steps significantly reduces target model invocations, improving speed but compromising generation quality. In contrast, configuration (c) with larger warmup steps maintains higher quality output but with reduced acceleration. To balance this trade-off, we divided the warmup phase into two distinct components (Stage-0 and Stage-1): first allowing the target model to establish critical structural elements, then employing small speculative steps to accelerate the remaining warmup procedure. As demonstrated in configuration (d), which

uses 2 steps for Stage-0 followed by speculative steps with $K = 3$ for Stage-1, then employing the same $K = 9$ speculative steps for Stage-2 as configurations (a) and (c). This multi-stage approach achieves nearly $2.1\times$ acceleration over the vanilla diffusion process while preserving better visual quality compared to configuration (a).

| Speedup: 2.4x | Speedup: 1x | Speedup: 1.9x | **Speedup: 2.1x** |

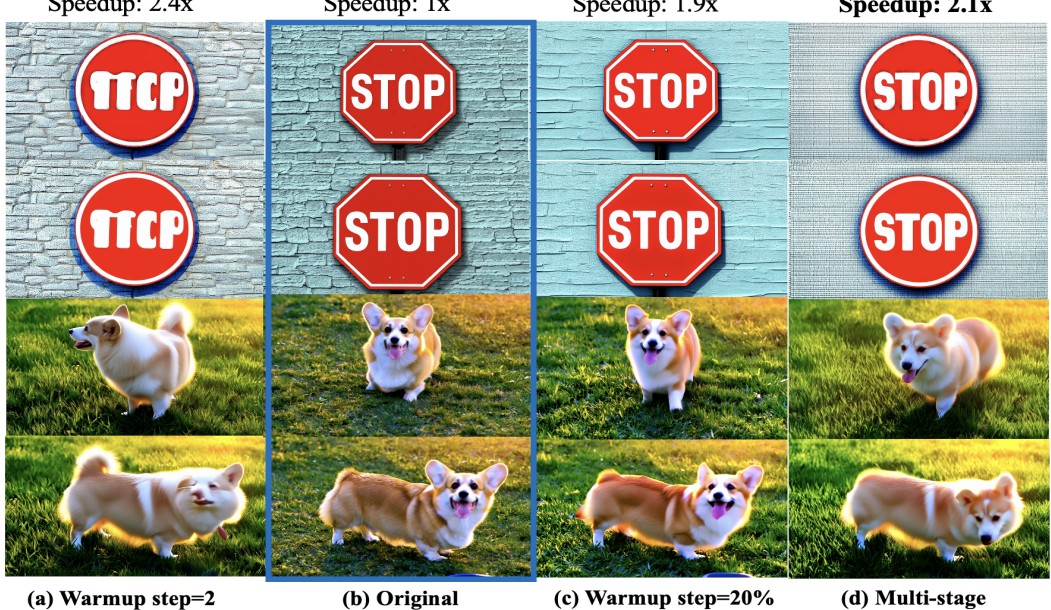

| (a) Warmup step=2 | (b) Original | (c) Warmup step=20% | (d) Multi-stage |

Figure 9: Comparison of different warmup and stage strategies: (a) minimal warmup with 2 steps followed by ASDSV Stage-2, (b) vanilla diffusion process, (c) conservative warmup with 20% of total steps (10 steps) followed by ASDSV Stage-2, (d) proposed multi-stage strategy with 2 steps for Stage-0 and 8 steps for Stage-1 (20% total diffusion steps as warmup), followed by the same ASDSV Stage-2.

## A.4 Results of FID against Real Images

ASDSV aims to preserve the generation quality of original models, so that we only present the FID compared with the original generationed images in the paper. We now provide the FID scores against real images on COCO Captions 2014 in Table 9:

Table 9: FID scores against real images on COCO Captions 2014.

| Method | FID (based ground truth) | FID (based original) ↓ |
|---|---|---|
| original | 35.63 | – |
| ASDSV-fast | **35.25** | 4.11 |
| ASDSV-slow | 35.31 | **3.90** |

The scores based ground truth here are very close, with a change of only 0.9%. Although the score for ASDSV-fast is higher (0.17% higher than ASDSV-slow), ASDSV-slow is actually closer to the original score, which also meets our expected goal.

## A.5 Expected Speedup of ASDSV

Here is the theoretical formula for the expected speedup, which incorporates the verification failure rate $E(\beta)$. The expected speedup factor for each stage is as follows:

$$S_{expected} = \frac{(1 - E(\beta))^2 \gamma + (1 - (1 - E(\beta))^2)}{\gamma c + 2} \tag{1}$$

**Proof.** Let the cost of running a single step of $M_p$ by $T$. An unoptimized round of ASDSV costs $\gamma cT + 2T$. This cost covers running the draft model $M_q$ $\gamma$ times and running the target model $M_p$

twice (at the first and last step). Assuming the two verification checks are independent, the expected number of steps advanced per cycle is given by:

$$E[\text{steps}] = (1 - E(\beta))^2 \gamma + (1 - (1 - E(\beta))^2) \cdot 1$$

This calculation assumes simple failure handling, where a failed verification results in advancing a single step. Thus, the effective cost per advanced step is the total cycle cost divided by the expected steps advanced:

$$\frac{\gamma c T + 2T}{(1 - E(\beta))^2 \gamma + (1 - (1 - E(\beta))^2)}$$

Since the cost of generating a single step with the target model $M_p$ is $T$, we get the desired result as equation (1).

As stated in Section 3.3, $c$ is a system-dependent constant, determined by the specific hardware and software implementation. A smaller c (a faster draft model) reduces the drafting overhead. However, it may imply a simpler model that struggles to approximate the target model's output, thus increasing the verification failure rate $E(\beta)$. In our experiments, conservatively-chosen $\gamma$ leads to few failure cases.

## B  Hyperparameter Auto-Tuning Process

As a follow-up to the sensitivity analysis in Section 4.3, this section details the principled, semi-automated process for determining the key hyperparameters for new models. We address each hyperparameter below:

**For $\gamma_1$ and Initial Steps $N$ (Generalization)**: We found $\gamma_1$ to be a less sensitive hyperparameter for the short Stage 1, generalizing well across models. A fixed, conservative value (typically 2 or 3) is a robust choice. The initial step count N is set to 5, analogous to the warm-up phase in other methods [10, 20].

**For the Stage Boundary (Profile Model Dynamics)**: We determine the boundary between Stage 1 and Stage 2 via a small-scale profiling run. By generating a small number of samples (as in Figure 2(a)), we can observe the model's output dynamics. This empirical data is crucial for setting the stage boundary.

**For $\gamma_2$ and $\delta$ (Iterative Tuning)**: We start with a candidate $\gamma_2$ based on the theoretical speedup ceiling for our target (Section 3.3). Next, using the verification loss curve obtained from the profiling stage (e.g., Figure 2(b)(c)), we set a corresponding threshold $\delta$ slightly above the observed loss at the $\gamma_2$-th step. Finally, we iteratively increase $\gamma_2$ and adjust $\delta$, evaluating the trade-off on a validation set until the desired speedup is met with minimal quality impact.

## C  Social Impact

The acceleration of diffusion models provided by ASDSV reduces computational resources and latency, improving real-time applicability of state-of-the-art diffusion models while promoting environmental sustainability through reduced energy consumption. However, it is important to note that ASDSV focuses primarily on efficiency gains and does not address inherent challenges such as privacy, bias, and fairness in the underlying diffusion models.

## D  Additional Visualization

We provide comprehensive visual comparisons between ASDSV and baseline acceleration methods for both text-to-image and text-to-video generation, as shown in Figure 10 and Figure 11. Extensive results demonstrate the superior visual fidelity of ASDSV across different generation scenarios.

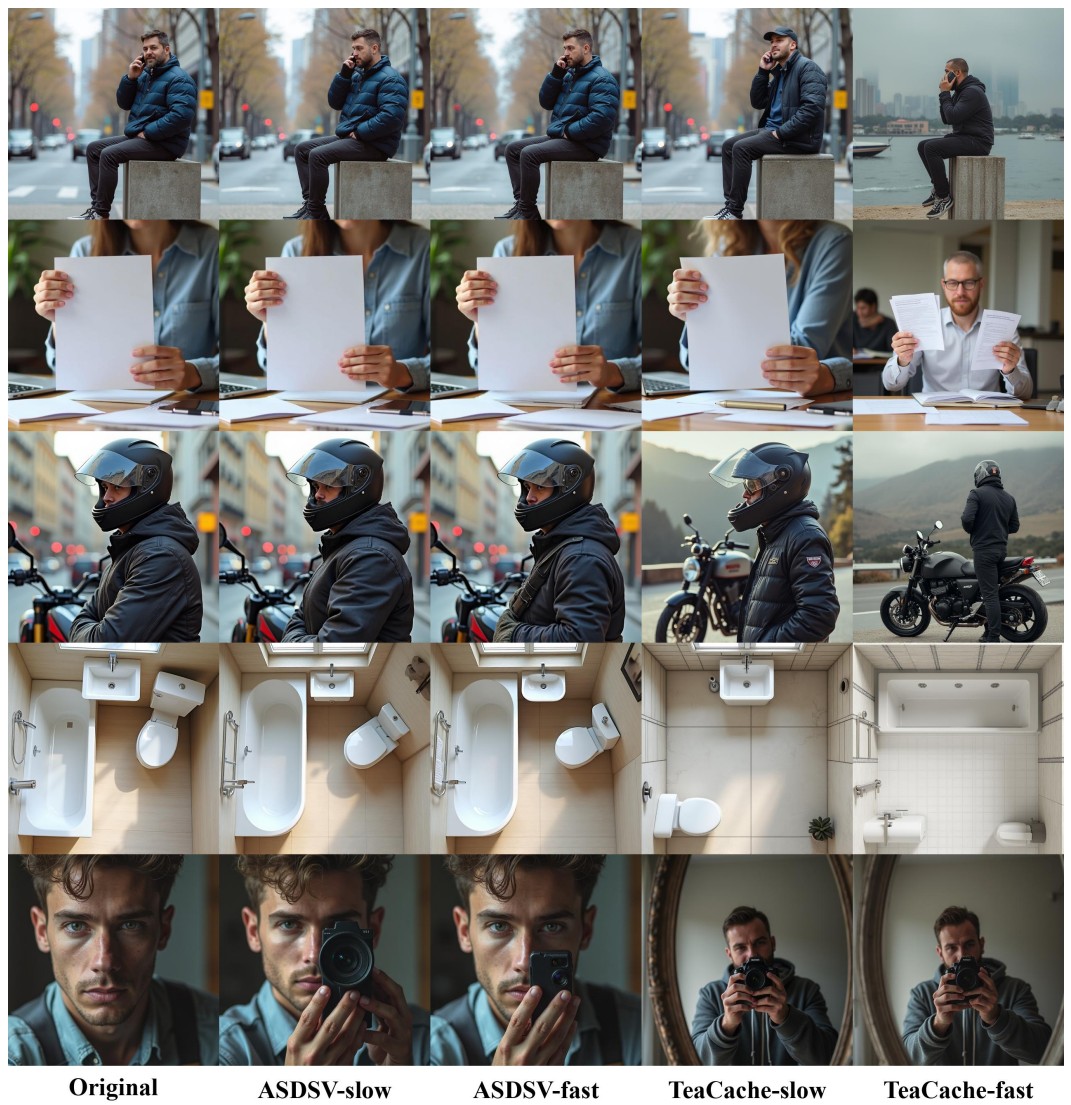

| Original | ASDSV-slow | ASDSV-fast | TeaCache-slow | TeaCache-fast |

Figure 10: Comparison of different accelerating methods on text-to-image generation using Flux.1-dev at 512x512 resolution.

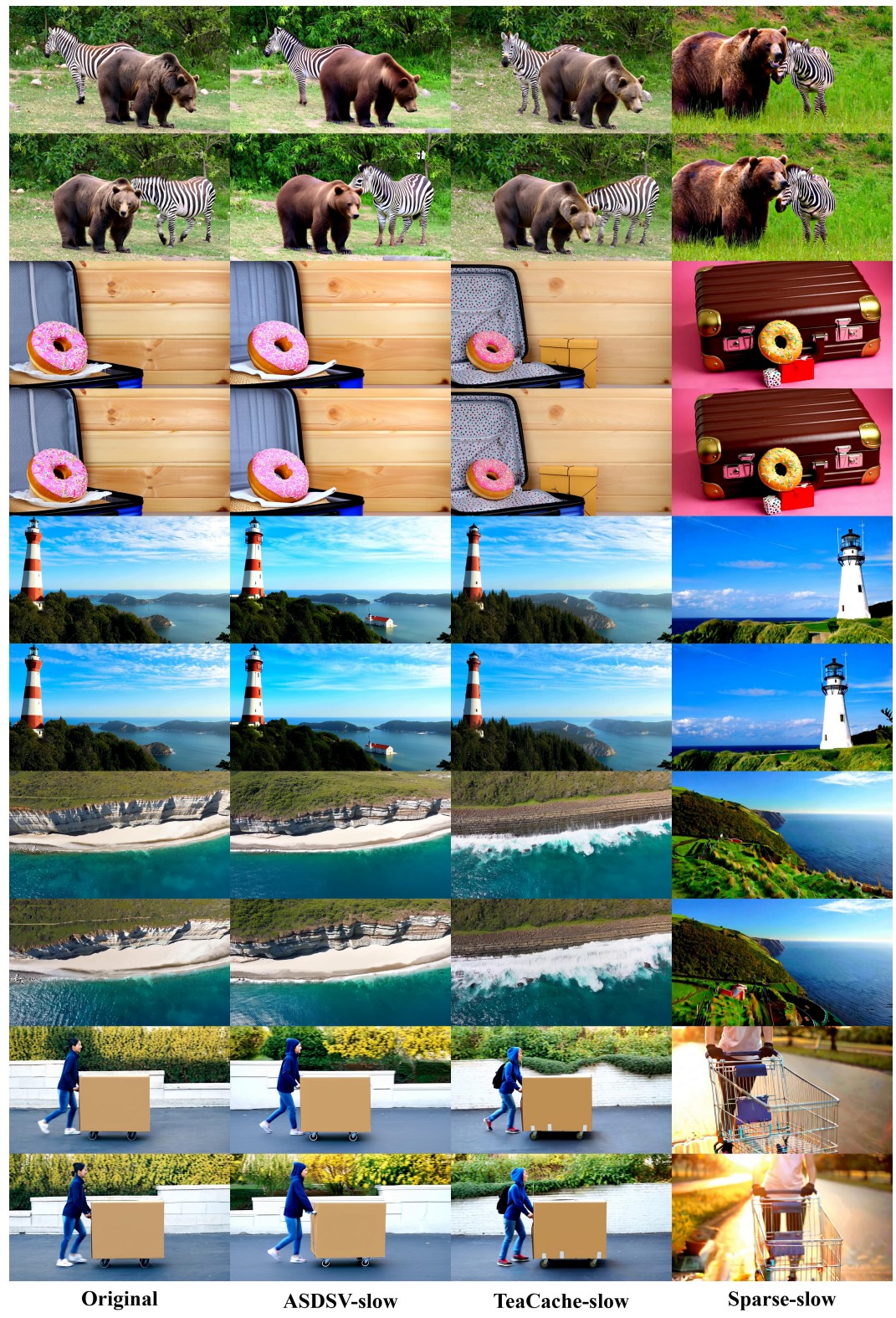

| Original | ASDSV-slow | TeaCache-slow | Sparse-slow |

Figure 11: Comparison of different accelerating methods on text-to-video generation using Wan2.1 at 480P resolution with 81 frames.

