# OpenReview forum: "ASDSV: Multimodal Generation Made Efficient with Approximate Speculative Diffusion and Speculative Verification"
_NeurIPS.cc/2025/Conference — NeurIPS 2025 poster_

### Official Review · Reviewer_yuAv · 2025-06-30

**Clarity:** 2
**Significance:** 1
**Originality:** 2
**Rating:** 3
**Confidence:** 5

**Summary:**

This paper proposes a method named SDSV, which aims to accelerate the inference of high-quality diffusion models (e.g., Flux and Wan2.1) by leveraging a lightweight "draft" model. The authors claim inspiration from the field of Speculative Decoding in Large Language Models (LLMs). The method's core idea is to use a "speculative verification" mechanism, which checks only the first and last steps of a generated sequence to decide on acceptance, combined with a multi-stage strategy to dynamically adjust the speculation length.

**Questions:**

* The Rigor of the Theoretical Analysis: Can you provide a more complete theoretical model for the expected speedup that incorporates the verification failure rate (as a function of δ and K), rather than assuming 100% success? How would this more realistic model change your discussion on the importance of the cost coefficient (c)?
* The Sensitivity of the Core Hyperparameter δ: Your paper only provides fixed values for δ for specific tasks. The choice of δ directly controls the trade-off between approximation error and acceleration efficiency, making it the central balancing point of your entire method. Can you please provide a sensitivity analysis for δ? For instance, how do the model's quality metrics (e.g., FID) and the measured speedup change as δ is varied from 0 to a larger value? This is critical for understanding the true performance envelope of your method.
* About writing: All formulas in the article are unnumbered.

**Ethical Concerns:**

["NO or VERY MINOR ethics concerns only"]

**Final Justification:**

The authors' rebuttal addresses my main concerns. Their method shows a clear advantage in balancing speed and quality, tackling the important problem of diffusion model acceleration.

However, the original manuscript had significant flaws in its theoretical framing and rigor, particularly regarding the misleading use of terminology (framing an approximative method as "speculative" in the lossless sense) and a lack of theoretical rigor in its analysis. Therefore, while I am raising my score based on their promised rebuttal, my recommendation remains cautiously optimistic. The paper's final quality is contingent on the authors thoroughly implementing the proposed changes, particularly renaming the method and strengthening the theoretical analysis.

**Limitations:**

Please see Weaknesses and Questions.

**Paper Formatting Concerns:**

No.

**Quality:**

1

**Strengths And Weaknesses:**

## Strengths
* Engineering Effectiveness: It is undeniable that the proposed set of techniques, combining a draft model, partial verification, and a multi-stage schedule, achieves significant speedups in practice. As an acceleration heuristic, the method demonstrates that it can keep visual quality degradation to a minimum under specific hyperparameter settings, which is a valuable reference.

## Weaknesses
* Fundamental Misuse of the "Speculative" Concept: This is the most critical flaw of the paper. In the LLM domain, speculative decoding has an unshakeable theoretical cornerstone: it is an unbiased acceleration technique that mathematically guarantees that its final output sequence has the exact same probability distribution as the sequence generated by the original target model autoregressively. It is a form of lossless speedup. The proposed SDSV, however, does not possess this property at all.
The core of SDSV relies on an L1 loss and a threshold δ to determine the acceptance of the draft model's output. This means its acceptance criterion is approximative. The distribution of its generated results will inevitably deviate from the original target model's distribution; it is, by definition, a form of lossy approximation.
The paper's own experiments, which repeatedly measure and report degradations in quality metrics (e.g., in FID, PSNR, and a 0.3%-0.4% drop in VBench score ), serve as proof against its own "speculative" framing. If the method truly guaranteed distributional identity, there would be no quality drop to measure, only a change in speed.

Therefore, naming the method "Speculative Diffusion" is highly misleading. It incorrectly implies that the method possesses the same theoretical guarantees as speculative decoding in the LLM field, which is an abuse of a core concept.

* Superficial Theoretical Analysis: The paper's theoretical analysis of the speedup ratio is also problematic.
The speedup formula presented in Section 3.3 is derived under the unrealistic assumption that "all verifications are successful". This completely ignores the most critical variable in the entire method: the verification failure rate, which is governed by the threshold δ and the step length K.
In true speculative decoding, the cost coefficient (c) and the acceptance rate are the two central factors that determine the final expected speedup. This paper's analysis fixes the acceptance rate at 100%, rendering the entire analysis superficial and incapable of revealing the method's true performance dynamics. A more rigorous analysis would model the expected speedup, E[speedup], as a function of c, K, and δ.

* Limited Novelty Beyond Heuristics: If we strip away the misleading "speculative" label, the method's core idea can be seen as "using a proxy model to generate candidates, then using a metric (L1 loss) to decide whether to skip the original model's computation." This is conceptually similar to existing cache-based skipping methods. For instance, TeaCache also uses a threshold to determine if adjacent inputs are similar enough to reuse a cache (i.e., skip a step). SDSV merely changes the comparison from "adjacent inputs" to "draft vs. target outputs" and the action from "reusing a cache" to "accepting the draft result." While the implementation is more complex, the novelty of the core concept—"threshold-based approximate skipping"—is questionable

---

> ### Author Rebuttal · Authors · 2025-07-31
>
> We appreciate the reviewer for the insightful and constructive comments. We respond to your comments as follows and sincerely hope that our rebuttal could properly address your concerns. If so, we would deeply appreciate it if you could raise your score. If not, please let us know your further concerns, and we will continue actively responding to your comments and improving our paper.
>
> ***W1:***  ***Misuse of the "Speculative" Concept. It incorrectly implies that the method possesses the same theoretical guarantees as speculative decoding in the*** ***LLM*** ***field.***
>
> We do apologize for causing confusion. Please let us clarify our original intent: 1) We never try to imply our method is lossless: we presented the quality drop in abstract, introduction, and evaluation in the submission. 2) We deliberately add "Speculative" before "Verification", with the intent to explain our verification uses an "approximate" way.
>
> In the revised version, we will emphasize the difference with speculative decoding in the LLM field at the beginning of the paper and use the name of "Approximate Speculative Diffusion with Speculative Verification (ASDSV)".
>
> ***W2/Q1: The Rigor of the Theoretical Analysis：1) Can you provide a more complete theoretical model for the expected speedup that incorporates the verification*** ***failure rate*** ***(as a function of δ and K)? 2) How would this more realistic model change your discussion on the importance of the cost coefficient (c)?***
>
> 1\) Yes. Here is a more complete formula for the expected speedup, which incorporates the verification failure rate $E(\beta)$. The expected speedup factor for each stage is as follows:
>
> $S_{expected} = \frac{(1-E(\beta))^2\gamma + (1 - (1-E(\beta))^2)}{\gamma c + 2} \tag{1} $
>
> **Proof.** Let the cost of running a single step of $M_p$ by $T$. An unoptimized round of SDSV costs $\gamma cT+2T$.  This cost covers running the draft model $M_q$ $\gamma$ times and running the target model $M_p$ twice (at the first and last step). Assuming the two verification checks are independent, the expected number of steps advanced per cycle is given by: $E[\text{steps}] = (1-E(\beta))^2 \gamma + (1 - (1-E(\beta))^2) \cdot 1$ This calculation assumes simple failure handling, where a failed verification results in advancing a single step. Thus, the effective cost per advanced step is the total cycle cost divided by the expected steps advanced: $\frac{\gamma cT+2T}{(1-E(\beta))^2\gamma + (1 - (1-E(\beta))^2)}$ Since the cost of generating a single step with the target model $M_p$is $T$, we get the desired result.
>
> 2\)  As stated in our submission, $c$ is still a system-dependent constant, determined by the specific hardware and software implementation [19].  A smaller c (a faster draft model) reduces the drafting overhead. However, it may imply a simpler model that struggles to approximate the target model's output, thus increasing the verification failure rate $E(\beta)$. In our experiments, conservatively-chosen $\gamma$ leads to few failure cases.
>
> We will add this analysis to Section 3.3 of our revised manuscript. Thank you for helping us strengthen our theoretical discussion.
>
> ***Q2: The Sensitivity of the Core*** ***Hyperparameter*** ***δ***
>
> Both $\gamma_1$, $\gamma_2$ depend on the temporal correlation between draft and target model on different stages, so SDSV is sensitive to these parameters (see Figure 2). Nevertheless, since the temporal correlation can be profiled offline, we can systematically search for the best candidates for these hyperparameters for each model offline. Please check the detailed analysis below and we will add them to the revised version.
>
> 1. **Analysis of $\gamma_1$ and its search method**
>
> First, we use brute force to search $\gamma_1$ because the space is small: it can only be an integer in (1,5). Below is the search space of Wan2.1. We only present one model result for simplicity, other models are the same.
>
> | $\gamma_1$ | LPIPS↓ | PSNR↑ | SSIM↑ | Speedup↑ |
> | :---: | :---: | :---: | :---: | :---: |
> | 2 | 0.23 | 18.57 | 0.61 | 1.70x |
> | 3 | 0.30 | 16.80 | 0.59 | 1.71x |
> | 4 | 0.30 | 16.76 | 0.59 | 1.72x |
>
>
> 2. **Analysis of $\gamma_2$ and its search method**
>
> Because the search space of $\gamma_2$ is larger than $\gamma_1$ (an integer selected from the range (1,15)), we developed a two-step iterative-guided search process for $\gamma_2$ to balance the speed-fidelity trade-off:
>
> - **Profile Model Dynamics:** We conduct a small-scale sampling run to profile the model's behavior, generating plots similar to Figure 2. This allows us to observe the output change dynamics and step-wise verification loss, which are crucial for setting the $\gamma_2$ and deciding the verification threshold.
> - **Iteratively Tune $\gamma_2$ and $\delta$:** We start with a candidate $\gamma_2$ based on the theoretical speedup ceiling for our target (Section 3.3). Next, using the verification loss curve (like Fig. 2b/c), we set a corresponding threshold $\delta$ above the observed loss at the $\gamma_2$-th step. We then incrementally increase $\gamma_2$ and adjust $\delta$, evaluating the trade-off on a validation set until the desired speedup is met with minimal quality impact.
>
> The following results show typical search results on Wan2.1 model, others are the same:
>
> | $\gamma_2$ | $\delta$ | LPIPS↓ | PSNR↑ | SSIM↑ | Speedup↑ |
> | :---: | :---: | :---: | :---: | :---: | :---: |
> | 3 | 0.1 | 0.24 | 18.89 | 0.60 | 1.2x |
> | 9 | 0.2 | 0.23 | 18.57 | 0.61 | 1.7x |
> | 12 | $\infty$ | 0.25 | 18.47 | 0.60 | 3.1x |
>
> The baseline for these comparisons is the original model, corresponding to the setting $\gamma_1=0, \gamma_2=0$. The setting of $\delta=\infty$ represents an "always-accept" strategy that effectively bypasses the verification stage to achieve maximum acceleration, corresponding to our SDSV-fast variant in Figure 5.
>
> ***W3:  Limited Novelty Beyond Heuristics. It is conceptually similar to existing cache-based skipping methods.***
>
> First, SDSV continues the line of using a fast path to accelerate multi-modal generation but introduces a new paradigm for the "fast path" in multi-modal generation: approximate speculative generation. Compared with caching (e.g., PAB [45], Teacache [24]), the fast path is augmented with a draft model to significantly enhance the quality and fidelity (as shown in Figures 1 and 5). Compared with non-approximate speculative (e.g., Traditional speculative diffusion [2]), the generation latency speedup is improved by 2x with negligible quality drop thanks to the significantly reduced verification overheads.
>
> Second, the design of an approximative speculation framework is far more challenging than simply deciding a threshold for skipping. As far as we know, we are the first to show that the vanilla draft-then-verify, which was designed for LLM, does not work well for SOTA, high-resolution multi-modal generation. Instead, we propose a draft-then-approximate-verify paradigm. It requires us to address which verification steps can be skipped, and how to balance the speed-fidelity trade-off. We propose systematic solutions including a multi-stage verification strategy and a system-level batched pipeline (deciding threshold is just one component of our overall design), and present promising results (e.g., 1.8-3X speedup with a minimal 0.3%-0.4% drop in VBench score) on diverse tasks using SOTA models like FLUX.1-dev and Wan2.1.
>
> ***Q3:  About writing: All formulas in the article are unnumbered.***
>
> Thanks for your suggestion. We will fix it in the revised version.

---

> > ### Comment · Reviewer_yuAv · 2025-08-04
> >
> > I've go through the authers rebuttal. The rebuttal is good, which I think the quality is better than the manuscript. : )

---

> > > ### Author Response · Authors · 2025-08-04
> > >
> > > Thank you for your response. We appreciate your valuable time and effort. If there are further concerns or questions, we are welcome to address them.

---

### Official Review · Reviewer_1KPp · 2025-07-02

**Clarity:** 3
**Significance:** 2
**Originality:** 2
**Rating:** 4
**Confidence:** 3

**Summary:**

This paper introduces Speculative Diffusion with Speculative Verification (SDSV), a novel method for accelerating multimodal diffusion models. It employs a lightweight "draft" model to propose future steps, which are then checked by the larger "target" model. The first innovation is "speculative verification" which cuts down the computational costs by only verifying the first and final outputs of a speculative sequence. Secondly, SDSV uses a multi-stage strategy that adjusts the number of speculative steps, employing fewer steps in the early stages and more in the stable later phases to optimize the speed-quality trade-off. On SOTA models like Flux.1-dev and Wan2.1, this approach achieves up to a 3.01x inference speedup and only 0.3%-0.4% drop in VBench quality scores.

**Questions:**

1. The hyperparameters are empirically chosen for specific model pairs. How well do they generalize to new models? Are there simpler, automated methods for tuning them?

2. Can you provide a direct cost comparison against traditional method (Speculative Diffusion) on a high-resolution task?

**Ethical Concerns:**

["NO or VERY MINOR ethics concerns only"]

**Final Justification:**

Thanks for the rebuttal. My concerns have been well addressed, and I will keep my positive score to accept this paper.

**Limitations:**

Yes

**Quality:**

3

**Strengths And Weaknesses:**

# Strengths:

1. Adequate experiments and compelling results: The method is supported by strong empirical evidence. The experiments are adequate, evaluating on SOTA models for both image and video generation and comparing against strong baselines like TeaCache and Sparse VideoGen. The speedups are substantial and the quality degradation is minimal.

2. Easy-for-implementation: SDSV is a plug-and-play inference optimization without the need to retrain or modify the original model architecture, which makes it highly practical.

# Weaknesses

1. Simple failure handling: The current strategy for handling a verification failure is to discard all subsequent speculative steps and revert to the target model. This approach is inefficient, especially when using a large number of speculative steps, as it wastes the computation from the draft model.

2. Complexity of Hyperparameter selection: The overall workflow needs many hyperparameters including the initial steps N, speculative steps for Stage-1 and Stage-2, and the number of rounds for each stage. The paper states these are chosen based on step-metric results (as in Fig 2b, 2c), which means that the tuning process is needed for every new model pair.

3. Lack of comparison with traditional Speculative Methods: The paper states that prior work's (Speculative Diffusion) verification process is too costly for high-resolution tasks. But a specific comparison in computation resources or time is missing.

---

> ### Author Rebuttal · Authors · 2025-07-31
>
> We thank the reviewer for the insightful and valuable comments. We especially appreciate the recognition that our work is a pratical solution and provides a thorough evaluation. We respond to your comments as follows and sincerely hope that our rebuttal could properly address your concerns.
>
> ***W1: Simple failure handling*** ***may waste the computation from the draft model.***
>
> As discussed in Section 5,  a binary search can be employed instead of discarding the entire K-step sequence. When a failure is detected at the final step, SDSV would recursively check the mid-step of the sequence to efficiently find the largest valid prefix of accepted steps. Since of our conservative hyperparameter setting, we encounter few failures and thus currently do not employ such a failure handling method.
>
> ***W2/Q1: The hyperparameters are empirically chosen for specific model pairs. How well do they generalize to new models? Are there automated methods for*** ***tuning*** ***them?***
>
> Yes, our hyperparameters can be determined for new models through an automated process. We address each parameter below:
>
> - **For $\gamma_1$ and Initial Steps $N$ (Generalization):** We found $\gamma_1$ to be a less sensitive hyperparameter for the short Stage 1, generalizing well across models. A fixed, conservative value (typically 2 or 3) is a robust choice. The initial step count N is set to 5, analogous to the warm-up phase in other methods [10,20].
> - **For the Stage Boundary (Profile Model Dynamics):** We determine the boundary between Stage 1 and Stage 2 via a small-scale profiling run. By generating a small number of samples (as in Fig. 2a), we can observe the model's output dynamics. This empirical data is crucial for setting the stage boundary.
> - **For $\gamma_2$ and $\delta$ (Iterative** **Tuning**):  We start with a candidate $\gamma_2$ based on the theoretical speedup ceiling for our target (Section 3.3). Next, using the verification loss curve obtained from the profiling stage (e.g., Fig. 2b/c), we set a corresponding threshold $\delta$ slightly above the observed loss at the $\gamma_2$-th step. Finally, we iteratively increase $\gamma_2$ and adjust $\delta$, evaluating the trade-off on a validation set until the desired speedup is met with minimal quality impact.
>
> ***W3/Q2: Can you provide a direct cost comparison against traditional method (Speculative Diffusion) on a*** ***high-resolution*** ***task?***
>
> Yes. Here is one direct comparison. On the Wan2.1 (Text-to-Video) model, traditional Speculative Diffusion takes 1,120s compared to the original model's 930s, i.e., increases the inference latency by 20%.
>
> This slow down is because the core assumption of the original speculative diffusion (Unet)—efficient parallel verification of draft steps—breaks down on SOTA, compute-bound diffusion in transformer (DiT) models.

---

> > ### Comment · Reviewer_1KPp · 2025-08-03
> >
> > Thanks for your rebuttal. My concerns have been addressed, and I will keep my positive score.

---

> > > ### Author Response · Authors · 2025-08-04
> > >
> > > Thank you for the positive feedback. We appreciate your valuable time and effort.

---

### Official Review · Reviewer_e9nk · 2025-07-02

**Clarity:** 3
**Significance:** 3
**Originality:** 3
**Rating:** 5
**Confidence:** 4

**Summary:**

The paper introduces speculative decoding in LLMs to diffusion models and proposes Speculative Diffusion with Speculative Verification. The proposed method uses a smaller version of the target model as a draft model to speed up the inference. The proposed method speeds up the inference of the diffusion model.

**Questions:**

1. In the proposed method, for every K steps, only the first and last steps’ noises will be compared and verified. The last step takes the output of the (N+K-1)-th step of the draft model as inputs, how to guarantee that the target model samples every step from N+1 to N+K-1, and the output of the (N+K-1)-th step of the target model will be the same?
2. In Table 1, I think the FID should also be calculated against the COCO Captions 2014 dataset. Or any other real datasets. Only calculating FID with the images generated by the base model is not enough, because it only indicates how similar the generated images are to those of the original model. We also care about the fidelity of the generated images compared to the real images.

**Ethical Concerns:**

["NO or VERY MINOR ethics concerns only"]

**Final Justification:**

The rebuttal addressed my questions regarding the details of the proposed method. I will keep my rating to accept the paper.

**Limitations:**

Yes

**Quality:**

3

**Strengths And Weaknesses:**

Paper Strengths:
1. The paper introduces the idea of speculative decoding in LLMs to diffusion models, which is significant to the vision community
2. The paper is well-written and easy to follow.
3. The proposed method speeds up the inference and does not harm the visual quality of generated images.

Paper Weaknesses:
1. In the proposed method, for every K steps, only the first and last steps’ noises will be compared and verified. The last step takes the output of the (N+K-1)-th step of the draft model as inputs, how to guarantee that the target model samples every step from N+1 to N+K-1, and the output of the (N+K-1)-th step of the target model will be the same?
2. In Table 1, I think the FID should also be calculated against the COCO Captions 2014 dataset. Or any other real datasets. Only calculating FID with the images generated by the base model is not enough, because it only indicates how similar the generated images are to those of the original model. We also care about the fidelity of the generated images compared to the real images.

---

> ### Author Rebuttal · Authors · 2025-07-31
>
> We thank the reviewer for the insightful and valuable comments. We especially appreciate the recognition that our work addresses a significant problem and represents a promising approach. We respond to your comments as follows and sincerely hope that our rebuttal could properly address your concerns.
>
> ***W1/Q1: In the proposed method, for every K steps, only the first and last steps’ noises will be compared and verified.How to guarantee that the target model samples every step from N+1 to N+K-1, and the output of the (N+K-1)-th step of the target model will be the same?***
>
> As detailed in Section 3.1, our speculative verification method chooses an approximative (threshold-based) while compution-less acceptance strategy instead of guaranteeing exactly the same output. We think this is a practical tradeoff between generation fidelity and latency.
>
> Specifically, as illustrated in Figure 2(a), after an initial warm-up phase, the step-wise change is highly consistent for draft and target models. That is, the magnitude of change between consecutive steps, measured by L1\_Loss becomes highly similar(i.e., $||D\_i - D\_{i-1}||\_1 \approx ||T\_i - T\_{i-1}||_1$). Based on  the draft and target model both trajectories evolve with a similar trend, if the start and last step of the current round are verified to be similar, and then the intermediate points are also highly likely to be valid (i.e., their verification loss would be below the threshold $\delta$).
>
> This high fidelity is demonstrated by the extensive quantitative metrics (e.g., FID, IS) and qualitative visual results presented in our evaluation.
>
> ***W2/Q2: Calculating*** ***FID*** ***between the generated images and the real images.***
>
> SDSV aims to preserve the generation quality of original models, so that we only present the FID compared with the original generationed images in the paper. Following your advice, we now provide the FID scores against real images on COCO Captions 2014:
>
> | Method | FID (based ground truth) | FID (based original) ↓ |
> | :---: | :---: | :---: |
> | original | 35.63 | -- |
> | SDSV-fast | **35.25** | 4.11 |
> | SDSV-slow | 35.31 | **3.90** |
>
> The scores based ground truth here are very close, with a change of only 0.9%. Although the score for SDSV-fast is higher (0.17% higher than SDSV-slow), SDSV-slow is actually closer to the original score, which also meets our expected goal.

---

### Official Review · Reviewer_9xwA · 2025-07-03

**Clarity:** 4
**Significance:** 4
**Originality:** 4
**Rating:** 5
**Confidence:** 2

**Summary:**

The authors propose to improve inference efficiency of diffusion models by adapting speculative decoding to diffusion transformers. They show their approach leads to significant speedup (up to 3x) with minimal drop in quality.

**Questions:**

- Can the authors include standard deviations over multiple seeds in their reported results?
- Can the authors comment on hyperparameter sensitivity to $\gamma_1$, $\gamma_2$?

**Ethical Concerns:**

["NO or VERY MINOR ethics concerns only"]

**Final Justification:**

I reserve my score of "accept"
- The authors provided extensive follow-up results that addressed my remaining concerns, in particular, statistic significance and hyperparamter sensitivity

**Limitations:**

yes

**Paper Formatting Concerns:**

no formatting concerns found

**Quality:**

4

**Strengths And Weaknesses:**

**Strengths**
- The proposed speculative decoding method leads to significant inference-time improvements (even under different video lengths and resolution) with minimal degradation in quality, which will have significant impact in the research community
- The proposed strategy of only verifying the first and last denoising steps is novel and impactful
- The authors provide extensive ablations and clear analysis of the theoretical speedup

**Weaknesses**
- Missing analysis statistical uncertainty. Error bars for quality-latency tradeoff in Figure 5 are not provided
- Missing analysis on hyperparameter sensitivity of $\gamma_1$, $\gamma_2$, which are manually selected

---

> ### Author Rebuttal · Authors · 2025-07-31
>
> We thank the reviewer for the insightful and valuable comments. We especially appreciate your recognition of an important problem, a novel method, and a clear analysis. We respond to your comments as follows and sincerely hope that our rebuttal could properly address your concerns.
>
> ***W1/Q1: Can the authors include standard deviations over multiple seeds in their reported results?***
>
> Our method is insensitive to the random seed. This is because the seed in the diffusion process only affects the initial noise, whereas the efficacy of SDSV depends on the temporal correlation between the draft and target models (as shown in Figure 2).
>
> To quantitatively verify this and address the reviewer's request, we further conducted a new set of focused experiments on three additional seeds (3461, 772190542, 118270042274490399) for our main comparisons (Original vs. SDSV) during the rebuttal period. Performance was consistent across both seeds (SDSV achieves better performance than other accerlation methods). The standard deviations over multiple seeds results on Text-to-Images over are below:
>
> | Images Metric | Ratio (Original / SDSV) |
> | :---: | :---: |
> | CLIP | 100.04% ± 0.10% |
> | IS | 103.84% ± 3.7% |
>
> Since most of the video metrics are reference-based, we cannot calculate a relative ratio (original/accelerated method). Therefore, we used a comparison between the accelerated methods (SDSV vs. Teacache), and SDSV is about 1.5x faster than Teacache.
>
> | Videos Metric | Ratio (SDSV / Teacache) |
> | :---: | :---: |
> | LPIPS↓ | 88.02% ± 2.85% |
> | PSNR↑ | 106.10% ± 1.25% |
> | SSIM↑ | 104.95% ± 1.84% |
>
> ***W2/Q2: Can the authors comment on*** ***hyperparameter*** ***sensitivity to $\gamma1, \gamma2$?***
>
> Both $\gamma_1$, $\gamma_2$ depend on the temporal correlation between draft and target model on different stages, so SDSV is sensitive to these parameters (see Figure 2). Nevertheless, since the temporal correlation can be profiled offline, we can systematically search for the best candidates for these hyperparameters for each model offline. Please check the detailed analysis below and we will add them to the revised version.
>
> 1. **Analysis of $\gamma_1$ and its search method**
>
> First, we use brute force to search $\gamma_1$ because the space is small: it can only be an integer in (1,5). Below is the search space of Wan2.1. We only present one model result for simplicity, other models are the same.
>
> | $\gamma_1$ | LPIPS↓ | PSNR↑ | SSIM↑ | Speedup↑ |
> | :---: | :---: | :---: | :---: | :---: |
> | 2 | 0.23 | 18.57 | 0.61 | 1.70x |
> | 3 | 0.30 | 16.80 | 0.59 | 1.71x |
> | 4 | 0.30 | 16.76 | 0.59 | 1.72x |
>
>
> 2. **Analysis of $\gamma_2$ and its search method**
>
> Because the search space of $\gamma_2$ is larger than $\gamma_1$ (an integer selected from range (1,15)), we developed a two-step iterative-guided search process for $\gamma_2$ to balance speed-fidelity trade-off:
>
> - **Profile Model Dynamics:** We conduct a small-scale sampling run to profile the model's behavior, generating plots similar to Figure 2. This allows us to observe the output change dynamics and step-wise verification loss, which are crucial for setting the $\gamma_2$ and decide the verfication threshold.
> - **Iteratively Tune $\gamma_2$ and $\delta$:** We start with a candidate $\gamma_2$ based on the theoretical speedup ceiling for our target (Section 3.3). Next, using the verification loss curve (like Fig. 2b/c), we set a corresponding threshold $\delta$ above the observed loss at the $\gamma_2$-th step. We then incrementally increase $\gamma_2$ and adjust $\delta$, evaluating the trade-off on a validation set until the desired speedup is met with minimal quality impact.
>
> The following results show typical search results on Wan2.1 model, others are the same:
>
> | $\gamma_2$ | $\delta$ | LPIPS↓ | PSNR↑ | SSIM↑ | Speedup↑ |
> | :---: | :---: | :---: | :---: | :---: | :---: |
> | 3 | 0.1 | 0.24 | 18.89 | 0.60 | 1.2x |
> | 9 | 0.2 | 0.23 | 18.57 | 0.61 | 1.7x |
> | 12 | $\infty$ | 0.25 | 18.47 | 0.60 | 3.1x |
>
> Due to the rebuttal format constraints, we are unable to show visual results here, but they will be included in the revised version.

---

> > ### Comment · Reviewer_9xwA · 2025-08-06
> > **Re: Rebuttal**
> >
> > Thank you for addressing my remaining concerns. I will keep my score of "accept".

---

> > > ### Author Response · Authors · 2025-08-07
> > >
> > > Thank you for the positive feedback. We appreciate your valuable time and effort.

---

### Official Review · Reviewer_C7bR · 2025-07-03

**Clarity:** 2
**Significance:** 3
**Originality:** 3
**Rating:** 4
**Confidence:** 4

**Summary:**

This paper introduces Speculative Diffusion with Speculative Verification (SDSV). The core idea is to leverage a compressed, lightweight model as a draft model in the diffusion process. To reduce the cost of verification, SDSV verifies only the initial and final states of the speculative steps. Furthermore, the authors divide the denoising process into multiple stages, assigning different speculative steps to each stage to enhance sample quality. Experiments on state-of-the-art models, FLUX and WAN, demonstrate the effectiveness of SDSV.

**Questions:**

What draft models are used in Table 1?

**Ethical Concerns:**

["NO or VERY MINOR ethics concerns only"]

**Final Justification:**

The rebuttal addressed my concerns about the paper. I will keep my rating to accept this paper.

**Limitations:**

Yes.

**Paper Formatting Concerns:**

No.

**Quality:**

3

**Strengths And Weaknesses:**

## Strengths

- The idea of using off-the-shelf compressed model as a draft model is interesting. It is also intuitive and easy to understand.
- The method is validated on leading image and video generation models (FLUX and WAN), demonstrating strong empirical performance.


## Weaknesses

- As acknowledged in the limitations, the method requires an additional draft model, which may not always be readily available. Acquiring such models can require significant effort. Additionally, using an extra model increases memory consumption.
- SDSV does not generalize well to low-step diffusion models; all experimental results are based on 50-step settings.
- The paper's writing needs to be improved:
  - Some mathematical notations are undefined prior to use—for instance, $D$, $T$, and $i$ in Line 131.
  - Figure 2’s plots are unclear. What is the "output trend" referred to in Line 134? How is it defined mathematically?
  - Lines 182–187: The intuition behind choosing $\gamma_2 = 9$ from Figure 2(b)(c) is not clear to me.

---

> ### Author Rebuttal · Authors · 2025-07-31
>
> We thank the reviewer for the insightful and valuable comments. We especially appreciate the recognition that our work addresses an important/interesting problem. We respond to your comments as follows and sincerely hope that our rebuttal could properly address your concerns.
>
> ***Q1. What draft models are used in Table 1?***
>
> The draft model used in Table 1 is Flux-SVD. As we described in Lines 232-233, Flux-SVD is the Flux.1-dev model quantized by the SVDQuant [21]. We will explicitly label "Flux-SVD" as the draft model in the caption of Table 1 in the revised version.
>
>
> ***W1. Using an extra model increases memory consumption.***
>
> Yes, but it can be alleviated. We measured the peak memory overhead to be +9GB (29.7%) for Text-to-Image and +5GB (8.0%) for Text-to-Video. However, the target and draft models do not need to run concurrently. This allows the extra memory consumption to be almost entirely eliminated by using asynchronous swapping without affecting inference latency (i.e., swapping the inactive model's parameters between GPU and CPU).
>
> ***W2. SDSV does not generalize well to low-step diffusion models; all experimental results are based on 50-step settings.***
>
> SDSV focuses on accelerating diffusion models which have more steps and thus face the issue of slow inference. Low-step diffusion models are out-of-scope. We will clarify the positioning of our work.
>
> Our experiments are based on the 50-step setting because most SOTA text-to-image and text-to-video models operate in this setting to ensure high-fidelity results. For example, leading models like FLUX.1-dev (T2I), Wan2.1 (T2V), and CogVideoX1.5 (I2V) all specify 50 steps as their default configurations.
>
> ***W3. The paper's writing needs to be improved.***
>
> Thanks for the detailed feedback. We will further proofreed our manuscript.
>
> 1. We will define all key mathematical symbols at their first appearance. For example, we will state that "$D_i$ and $T_i$ denote the predicted outputs from the draft and target models at step $i$, respectively," ensuring all notations are unambiguous throughout the paper.
>
> 2. We will provide a more precise definition for the term "output trend" in Line 134. This refers to our key empirical observation: the strong temporal correlation between draft and target models in the diffusion process. Specifically,  as shown in Figure 2(a), the magnitude of change of the draft model's output and the target model's output between consecutive steps is highly similar (i.e., $||D\_i - D\_{i-1}||\_1 \approx ||T\_i - T\_{i-1}|\|_1$) after the first $N$ initial steps.
>
> 3. We will add a detailed explanation for our choice of $\gamma_2$ in Line 183. As we now illustrate with a red vertical line in Figure 2(b)(c), the verification error curve exhibits a sharp increase at the 9th step. This inflection point shows that the draft model's outputs are starting to diverge significantly from the target model's. Therefore, it represents the appropriate moment to halt the drafting phase.

---

> > ### Comment · Reviewer_C7bR · 2025-08-03
> > **Re: Rebuttal**
> >
> > The rebuttal addressed my concerns about the paper. I will keep my rating to accept this paper.

---

> > > ### Author Response · Authors · 2025-08-04
> > >
> > > Thank you for the positive feedback. We appreciate your valuable time and effort.

---

### Note · Authors · 2025-08-12

# General Response

For clarity and simplicity, we will refer to Reviewers C7bR, 9xwA, e9nk, 1KPp and yuAv as R1, R2, R3, R4 and R5 respectively, in the following response.

We sincerely thank the reviewers and the Area Chair for their valuable time and effort. We appreciate that reviewers found our method "interesting, novel, and impactful" (R1, R2) and "easy to follow and understand" (R1, R3)**,** highlighting its "significant inference-time improvements with minimal quality degradation" (R2, R3)**.** Our work was also praised for its "engineering effectiveness" (R4, R5), and "strong empirical performance on SOTA diffusion models" based on "adequate experiments" (R1, R4).

We have summarized the rebuttal and our planned revisions below.

## Summary of Rebuttal

We have successfully addressed the concerns of the four initially positive reviewers: R2 and R3 (Rating: 5) as well as R1 and R4 (Rating: 4). R1, R2, and R4 have explicitly confirmed that our responses resolved their questions and will maintain their positive ratings. R3 also has not raised any new questions.

The only initial negative reviewer, R5, also stated that "the rebuttal is good".

## Summary of Planned Revisions

***Updates of results:***

- **Section-3.3**: Incorporating the theoretical analysis of expected speedup.
- **Section-4.3**: Adding the ablation study on hyperparameter $\gamma_1$ and $\gamma_2$ sensitivity and detailing our automated tuning process.
- **Appendix-A.3**:  Adding the multi-seed stability analysis with standard deviations and the evaluation with FID scores computed against real images.

***Refinements to writing:***

- Renaming our method to "ASDSV" (A for Approximate); Numbering all equations.
- **Section-3.1**:  Adding definitions for key symbols $D_i$, $T_i$, providing a more precise definition for the term "output trend" and explanation for our choice of $\gamma_2$.
- **Section-4.2**:  Labeling "Flux-SVD" as the draft model in Table 1.

We hope that our responses and planned revisions have successfully addressed the reviewers' primary concerns and will significantly strengthen our manuscript.

---

### Decision · Program_Chairs · 2025-09-17

**Decision:**

Accept (poster)

**Comment:**

This paper explores a new technical perspective for accelerating multimodal diffusion models: formulating a speculative verification strategy and a multi-stage speculative strategy, conditioned on a dual use of a draft model and a target model, inspired by speculative decoding. By these two strategies, the initial and final speculative outputs of K speculative steps by the draft and target model need to be aligned, facilitated by different-stage-value settings of K. The efficacy of the proposed method is validated on Flux.1-dev for image generation and Wan2.1 for video generation.

The paper initially|finally scored (4,5,5,4,2)|(4,5,5,4,3) by five knowledgeable reviewers, who mostly recognized the motivation, the idea and the basic performance of the proposed method. Meanwhile, the reviewers also raised some concerns about 1) improvements of the writing in some parts; 2) more experiments to study the generalization of the proposed method; 3) method limitations, e.g., for every K steps, only the first and last steps’ noises of the draft and target models will be verified, which is not optimal; 4) missing ablations, e.g., studying the choice of hyperparameters.

The authors provided detailed responses to these concerns, which were recognized by all of five reviewers. Finally, 4 reviewers maintained their positive scores, 1 reviewer (yuAv) increased the score from Reject to Borderline reject, criticizing the misleading use of word "speculative" in method formulation. The AC read the paper, the reviews, the rebuttal, the author-reviewer discussion and the reviewers' feedback, and agree with most of reviewers' assessment. Therefore, I recommend to accept this paper (poster). The authors are encouraged to carefully consider the reviewers' comments/suggestions and their rebuttal in the final paper revision.